# The Effect of Neddylation Blockade on Slug-Dependent Cancer Cell Migration Is Regulated by p53 Mutation Status

**DOI:** 10.3390/cancers13030531

**Published:** 2021-01-30

**Authors:** Yelee Kim, Jun Bum Park, Junji Fukuda, Masatoshi Watanabe, Yang-Sook Chun

**Affiliations:** 1Department of Biomedical Science, Seoul National University College of Medicine, Seoul 03080, Korea; whyj8007@snu.ac.kr (Y.K.); junbump@snu.ac.kr (J.B.P.); 2Ischemic/Hypoxic Disease Institute, Seoul National University College of Medicine, Seoul 03080, Korea; 3Faculty of Engineering, Yokohama National University, Yokohama 240-8501, Japan; fukuda@ynu.ac.jp; 4Oncologic Pathology, Graduate School of Medicine, Mie University, 2-174 Edobashi, Tsu 514-8507, Japan; mawata@doc.medic.mie-u.ac.jp; 5Department of Physiology, Seoul National University College of Medicine, Seoul 03080, Korea

**Keywords:** neddylation, p53, Slug, PI3K/Akt/mTOR signaling, migration, epithelial-mesenchymal transition (EMT)

## Abstract

**Simple Summary:**

Neddylation is a process in which the small ubiquitin-like molecule NEDD8 is covalently conjugated to target proteins by sequential enzymatic reactions. Because neddylation plays critical roles in regulating cancer growth and migration, it is emerging as an effective therapeutic target. The major tumor suppressor protein p53 reduces cancer cell migration and is inhibited by neddylation. As p53 is lost or mutated in 50% of various cancer types, this study attempted to investigate how neddylation affects cancer cell migration according to p53 status. Neddylation blockade reduced or caused no change in migration of wild type or mutant p53 cancer cell lines. In contrast, neddylation blockade induced migration of p53-null cancer cell lines. These results were mediated by the differential effect of neddylation blockade on the epithelial–mesenchymal transition activator Slug according to p53 status. Thus, the p53 status of cancer cells should be considered when developing neddylation-targeted anticancer drugs.

**Abstract:**

The tumor suppressor protein p53 is frequently inactivated in human malignancies, in which it is associated with cancer aggressiveness and metastasis. Because p53 is heavily involved in epithelial–mesenchymal transition (EMT), a primary step in cell migration, p53 regulation is important for preventing cancer metastasis. p53 function can be modulated by diverse post-translational modifications including neddylation, a reversible process that conjugates NEDD8 to target proteins and inhibits the transcriptional activity of p53. However, the role of p53 in cancer migration by neddylation has not been fully elucidated. In this study, we reported that neddylation blockade induces cell migration depending on p53 status, specifically via the EMT-promoting transcription factor Slug. In cancer cell lines expressing wild type p53, neddylation blockade increased the transcriptional activity of p53 and expression of its downstream genes p21 and MDM2, eventually promoting proteasomal degradation of Slug. In the absence of p53, neddylation blockade increased cell migration by activating the PI3K/Akt/mTOR/Slug signaling axis. Because mutant p53 was transcriptionally inactivated but maintained the ability to bind to Slug, neddylation blockade did not affect the migration of cells expressing mutant p53. Our findings highlight how the p53 expression status influences neddylation-mediated cell migration in multiple cancer cell lines via Slug.

## 1. Introduction

Cancer metastasis is a sequential process by which cancer cells spread from their original site to distant organs. Because many patients are diagnosed after cancer dissemination occurs, metastatic cancer remains a hallmark of malignancy among various cancer types [1]. Notably, small cell lung cancer metastasizes to diverse organs at the time of initial detection [2]. In addition, more than two-thirds of ovarian cancers are not detected until the late stages, in which cancer cells have spread to the peritoneal cavity [3]; the existing treatments for metastatic cancer (e.g., surgery, chemotherapy, and immunotherapy) are generally unable to cure the cancer [4]. Thus, metastatic cancer is the leading cause of cancer-related mortality [1] and elucidating the dynamics of cancer metastasis is regarded as a powerful strategy for effective cancer treatment.

Recently, a major tumor suppressor protein, p53, has been shown to participate in metastasis [5]. p53 is a transcription factor that facilitates expression of target genes that negatively regulate epithelial–mesenchymal transition (EMT) [6]. During metastasis, EMT leads to cell migration by converting epithelial cells into mesenchymal cells via loss of cell–cell adhesion and cell polarity, and the acquisition of invasive properties [7]. The EMT process involves EMT-related transcription factors such as Snail, Zeb1, Twist, and Slug [8]. These factors promote EMT by binding to E-box domain of E-cadherin promoter, a major epithelial gene, and repressing its expression [9]. p53 inhibits cancer EMT by positively regulating E-cadherin via various mechanisms. For instance, p53 represses the expression of Snail and Zeb1 by inducing miR-34 and miR-200, respectively [10,11]. Additionally, p53, p21, and MDM2 bind to Slug and promote its ubiquitin-mediated proteasomal degradation [12,13]. Thus, loss of p53 function promotes cell migration and tumor metastasis [14]. More than 50% of human cancers involve p53 deletion or a functional mutation in p53 [15]; therefore, a better understanding of the relationship between p53 disruption and EMT will help identify the cause of metastasis in affected cancers.

p53 function is regulated by multiple post-translational modifications such as ubiquitination, phosphorylation, sumoylation, and neddylation [16]. Importantly, protein neddylation is a process whereby a small ubiquitin like protein neuronal precursor cell-expressed developmentally downregulated protein 8 (NEDD8) is conjugated to target proteins via NEDD8 activating enzyme E1 (NAE1), NEDD8 conjugating enzyme E2, and NEDD8 E3 ligase [17]. Neddylation is strongly involved in cancer physiology. For instance, neddylation at the C-terminus of p53 (K370, K372, and K373) by MDM2 reduces its transcriptional activity [18]. Furthermore, NEDD8 conjugation of E2F1—a transcription factor related to cell proliferation and death—modifies its target gene specificity and regulates apoptosis [19]. Neddylation also has diverse influences on oncogenic signaling pathways such as PI3K/Akt/mTOR and Transforming Growth Factor-β (TGF- β) [20,21].

There is increasing evidence that neddylation is involved in cancer cell migration. Generally, neddylation blockade is presumed to inhibit cancer cell migration because neddylation is reportedly overactivated in some human cancers (e.g., hepatocellular carcinoma and glioblastoma) [22,23]. However, Zhou et al. recently suggested that neddylation blockade induced in vivo tumor proliferation and in vitro cancer cell migration [24]. In addition, our previous studies demonstrated that neddylation blockade enhanced cancer cell migration by stimulating various oncogenic pathways [20,25]. Despite the conflicting results concerning the role of neddylation in cancer cell migration, the underlying molecular mechanisms responsible for these effects remain unclear.

Given that neddylation blockade induces cell migration in variety of cancer cell lines, and p53 is profoundly involved in cancer cell EMT [5,20], we assumed that the p53 status may determine cancer cell migration via neddylation. By inducing neddylation blockade, we discovered the role of Slug as a mediator of cancer cell migration. MLN4924 (Pevonedistat), a neddylation inhibitor that specifically blocks NAE1, was able to induce cell migration in p53-null cancer cells by upregulating the PI3K/Akt/mTOR/Slug signaling axis. Conversely, in cancer cell lines expressing wild type p53, neddylation blockade induced p53 transcriptional activity and eventually promoted proteasomal degradation of Slug, which resulted in reduced or stable cell migration. However, mutant p53 still form complexes with MDM2, p21 and Slug, and resulted in no cell migration upon blocking neddylation. Taken together, the findings demonstrate the importance of p53 in determining the effects of neddylation blockade on cancer cell migration.

## 2. Materials and Methods

### 2.1. Cell Lines and Cell Culture

PA-1 and ES-2 cell lines were obtained from the American Type Culture Collection (ATCC; Manassas, VA, USA). A2780 cells were purchased from European Collection of Authenticated Cell Cultures (ECACC). MCF7, A549, SKOV-3, PC3, and H1299 cell lines were purchased from Korean Cell Line Bank (Seoul, Korea). MCF7, A549, A2780, SKOV-3, and ES-2 cells were cultured in RPMI1640. PC3 and H1299 were incubated in Dulbecco’s modified Eagle’s medium (DMEM). PA-1 cells were incubated in MEM. All media were supplemented with 10% FBS and 1% penicillin.

### 2.2. Antibodies and Chemicals

Slug (9585S), CUL1 (4995S), NEDD8 (2745S), p-AKT (ser473) (9271S), phospho-mTOR (ser2448) (2971S), E-Cadherin (3195S), Phospho-STAT3 (9145S), Phospho-Smad2/3 (8828S), Phospho-ERK (9101S), and antibodies of EMT markers (ZO-1, Vimentin, ZEB1, N-cadherin, Claudin-1, β-Catenin, Snail) (Epithelial–Mesenchymal Transition (EMT) Antibody Sampler Kit 9782T) were purchased form Cell Signaling Technology (Danvers, MA); β-tubulin (sc-166729), p53 (sc-126), p21 (sc-6246), MDM2 (sc-965), and Ubiquitin (sc-8017) antibodies from Santa Cruz Biotechnology (Dallas, TX). MLN4924 was synthesized, as previously described [26]. MG132 (BML-PI102) (Enzo Life Sciences, Plymouth Meeting, PA, USA), Cycloheximide (C1299) (Sigma Aldrich, St. Louis, MO, USA), LY294002 (L9908), MK-2206 (s1078), and Rapamycin (s1039) (Selleck Chemicals, Houston, TX, USA) were purchased from the indicated company.

### 2.3. Plasmids, siRNAs, and Transfection

Full-length wild type p53 cDNAs were inserted into Flag/Streptavidin binding protein (F/S)-tagged pcDNA. pcDNA vector was used as the negative control. For transient transfection of plasmids or siRNAs, cells about 70% confluency were transfected using Lipofectamine 2000 reagent (plasmid) or Lipofectamine RNAiMAX (siRNA) reagent. Control, NEDD8, Slug, p53 siRNAs were synthesized by M. Biotech (Hanam-si, Gyeonggi-do, Korea). The nucleotide sequences of siRNAs are as follows:

Human Control: 5′-UUGAGCAAUUCACGUUCAUTT-3′

Human NEDD8 (#1: 5′-AGCGGUAGGAGCAGCAAUUUAUCCG-3′

Human NEDD8 (#2): 5′-CCCUGGUUGUCAAUAAAAUAUUUCC-3′

Human SNAI2 (#1): 5′-CAAUAAGACCUAUUCAACUUUUUCT-3′

Human SNAI2 (#2): 5′-ACUGAGUGACGCAAUCAAUGUUUAC-3′

Human *TP53* (#1): 5′-AGCAUCUUAUCCGAGUGGAAGGAAA-3′

Human *TP53* (#2): 5′-GAGGUUGGCUCUGACUGUACCACCA-3′

### 2.4. Wound Healing Assay

Cells were seeded in 12-well plates and cultured until confluence. The cells were scratched with a 200 μL pipette tip, washed with serum-free media to eliminate debris, and incubated in serum-free media for 24 h. After incubation, migrated cells were acquired in three randomized fields at the lesion border by an inverted microscope. The area of migration was quantified using ImageJ software (National Institutes of Health, Bethesda, MD, USA).

### 2.5. Transwell Migration Assay

Cells (2–8 × 10^4^) were resuspended in 200 μL serum-free medium with designated concentration of reagents and seeded onto collagen coated Boyden chambers (Transwell^®^; Costar Corp., Cambridge, MA, USA; 6.5 mm diameter, 8 μm pore size). Then, 500 μL of complete medium was filled in the bottom chambers. Cells were allowed to migrate for 24 h. After clearing the non-invading cells on the inner surface of the chamber using a cotton swab, the outer surfaces were fixed in MeOH and stained with crystal violet for 40 min. Four independent areas per filter images were captured using an inverted microscope, and the number of cells that migrated was counted by ImageJ software.

### 2.6. Western Blotting and Immunoprecipitation

Cell lysates were separated on SDS-polyacrylamide gels and transferred to Immobilon-P membranes (Millipore, Bedford, MA, USA). Membranes were blocked with 3% skim milk or 3% bovine serum albumin (BSA) in Tris/saline solution containing 0.1% Tween-20 (TTBS) for 1 h, and incubated overnight at 4 ℃ with primary antibody (dilution, 1:500‒1:3000) in the respective blocking solution. The membranes were then incubated with a horseradish peroxidase-conjugated secondary antibody (1:2000) for 1h at room temperature, and visualized using the ECL Plus kit (Thermo Fisher Scientific, Waltham, MA, USA). To precipitate protein interactions, cells were lysed with buffer containing 5 mM EDTA, 50 mM Tris-Cl, 100 mM NaCl, 0.1% NP-40, and a protease inhibitor cocktail (Sigma-Aldrich). Cell lysates were then incubated with anti-Slug or anti-p53 overnight at 4 °C and with Protein A/G Sepharose beadsTM (GE Healthcare Life Sciences, Marlborough, MA, USA) at 4 °C for 4 h. After incubation, the precipitates were washed for three times. Then, the beads were eluted with 2× SDS buffer and subjected to western blotting (Appendix A).

### 2.7. RNA Isolation and RT-qPCR

Isolation of total RNA was conducted using TRIzol reagent (Invitrogen, Carlsbad, CA, USA). cDNA was synthesized and amplified using an EasyScript cDNA Synthesis Kit (Applied Biological Materials Inc., Richmond, BC, Canada). Then, cDNAs were measured with EvaGreen qPCR master mix reagent (Applied Biological Materials) using a StepOne Real-time PCR System (Applied Biosystems, Foster City, CA, USA). The sequences of the primer used in the experiments as follows—*18S*: forward 5′-TTCGTATTGAGCCGCTAGA-3′, reverse 5′-CTTTCGCTCTGGTCCGTCTT-3′; *SNAI2*: forward 5′-TGTTGCAGTGAGGGCAAGAA-3′, reverse 5′-GACCCTGGTTGCTTCAAGGA-3′; *MDM2*: forward 5′-CCGGATCTTGATGCTGGTGT-3′, reverse 5′-CTGATCCAACCAATCACCTGAAT-3′; *CDKN1A*: forward 5′-AGCGATGGAACTTCGACTTTG-3′, reverse 5′ CGAAGTCACCCTCCAGTGGT-3′; *TP53*: forward 5′-ATGGAGGAGCCGCAGTCAGA-3′, reverse 5′-AGTTGTCAGTCTGAGTCAGG-3′.

### 2.8. MTT Assay

To check cell viability, cells treated with MLN4924 (125 nM) for 24 h were incubated with 0.5 mg/mL of MTT (3-(4,5-dimethylthiazol-2-yl)-2,5-diphenyltetrazolium bromide) for 2 h at 37 °C, 5% CO_2_ atmospheric condition. The medium was removed and precipitated purple formazan was solubilized with 200 μL dimethyl sulfoxide, and quantified at 570 nm by spectrophotometry.

### 2.9. Luciferase Reporter Assay

Wild type p53 cells were co-transfected with PG13-luc (wt p53 binding sites) (Addgene) and β-gal plasmid. p53-null cells were introduced with PG13-luc (wt p53 binding sites), β-gal plasmid, and Wt-p53 DNA. After 24 h of incubation, vehicle or MLN4924 were treated for another 24 h and harvested. Luciferase activities were analyzed using a Lumat LB9507 luminometer (Berthold Technologies, Bad Wildbad, Germany); β-gal activities were used as control.

### 2.10. Immunofluorescence

Immunofluorescence was performed for cells placed in 12-well plates. Cells treated with MLN4924 for 24 h were fixed in 4% paraformaldehyde for 10 min and washed three times with PBS without permeabilization. The cells were then incubated in blocking solution (PBS + 0.1% Tween 20 + 3% bovine serum albumin) for 1 h. After blocking, the cells were stained with E-cadherin antibody (1:200 dilution) at 4 °C for overnight. Subsequently, cells were washed for three times with PBS and added with the secondary antibody (Alexa Flour 458, anti-rabbit, 1:500 dilution) for 1 h. Next, cells were washed three times with PBS and stained with DAPI (1:1000 dilution) for 1 min to visualized nuclei. Images were captured by a confocal microscope (FV3000, OLYMPUS, Tokyo, Japan).

### 2.11. D Spheroid Cell Culture

3D spheroid cell culture method is derived from Prof. Fukuda. PA-1 (1 × 10^6^), A2780 (5 × 10^5^), and SKOV-3 (6 × 10^5^) cells were seeded on PDMS plate coated with 4% Pluronic. Cells were incubated in the oxygen permeable plate for 5 days and average diameter was measured by Image J.
Roundness (%) = 100 − (R − r)/R × 100

(R = the radius of the minimum circumscribed circle and r = the radius of the maximum inscribed concentric circle).

### 2.12. Bioinformatics Analysis

Gene Expression Profiling Interactive Analysis (GEPIA) was used to investigate NAE1 expression between 27 human cancer types to their normal matches [27]. This webserver extracts mRNA expression data from 9736 tumors and 88,587 normal samples of TCGA (The Cancer Genome Atlas) data portal and GTEx (Genotype-tissue Expression) database of normal tissues (http://gepia.cancer-pku.cn).

Four independent ovarian cancer gene expression profile data, GSE14407, GSE26712, GSE40595, and GSE17308, were collected from the National Center for Biotechnology Information’s Gene Expression Omnibus (GEO, NCBI). GSE14407 included 12 normal ovarian surface epithelia (OSE) samples and 12 serous ovarian cancer epithelia samples; GSE26712 included 10 normal ovarian surface epithelium and 185 primary ovarian tumors from high; GSE40595 included 6 normal ovarian surface epithelium and 32 ovarian tumor epithelial component; and GSE17308 included 4 normal ovarian surface epithelia samples, 7 benign ovarian serous tumors, and 28 invasive ovarian serous tumors. We analyzed expression profiles of NAE1 (202268_s_at; corresponding to NAE1) in above datasets. In order to identify and compare the *TP53* somatic mutation prevalence among various cancer type, TCGA data were downloaded from the International Cancer Genome Consortium (ICGC) web portal (https://dcc.icgc.org/).

### 2.13. Statistical Analysis

All experiments were repeated at least three or more experiments. GraphPad Prism 8 software (GraphPad Inc., La Jolla, CA, USA) or Microsoft Excel 2013 software (Microsoft Corp., Redmond, WA, USA) were used to analyze all data. All data were presented as the means and standard deviation (SD). Two-tailed, Student’s *t*-test was conducted for general statistical analyses and Mann–Whitney U test was used for comparing protein levels. Statistical significance was considered at *p*-value less than 0.05.

## 3. Results

### 3.1. Neddylation Blockade Induces Cancer Cell Migration according to p53 Status

Given that neddylation blockade increases the migration of p53-null cancer cell lines [20,25,28], we investigated the relationship between p53 and neddylation blockade in cancer cell migration using different cancer cell lines with wild type p53 or p53-null characteristics (wild type p53: PA-1, MCF7, and A549 cells; p53-null: SKOV-3, PC3, and H1299 cells). Consistent with the results of Lee et al. [20], treatment with the NAE1 inhibitor MLN4924 (125 nM) for 24 h after incubation in serum-free medium for 1 day did not affect the number of cells (Appendix A). Because MLN4924 induced the p53-dependent DNA damage response and cell death [29], we conducted an MTT assay and confirmed that the optimized condition did not influence cell viability (Appendix A).

To examine the effects of neddylation blockade on cell migration, cancer cells were treated with MLN4924 and NEDD8-targeting siRNA (si-NEDD8) and then analyzed by wound healing and Transwell assays. The treatment of MLN4924 was confirmed by reduction of NEDD8-CUL1 and NEDD8-modified protein expression (Figure 1A and Appendix A). The results showed that treatment with MLN4924 significantly induced the migration of p53-null cells, whereas wild type p53 cell lines showed reduced, or no significant change in cell migration (Figure 1A,B). In addition, enhanced p53-null cancer cell migration by MLN4924 was inhibited by wild type p53 transient expression (Appendix A). Consistent with these findings, transfection with si-NEDD8 prevented migration of wild type p53 cell lines, whereas it enhanced migration of p53-null cancer cells (Figure 1C,D). These results were confirmed using different NEDD-targeting siRNA sequences (Appendix A). Collectively, neddylation blockade had opposing effects on cancer cell migration according to p53 status.

### 3.2. Associations of NAE1 Expression, the Pathological Stage, and TP53 Mutation Prevalence in Ovarian Tumors

Dysregulation of neddylation is associated with aggressiveness in various cancer types [22,30,31]. To investigate the expression pattern of NAE1 in patients with cancer, we used the Gene Expression Profiling Interactive Analysis (GEPIA) webserver to compare the expression level of NAE1 between tumor and normal tissues in 27 cancer types. In particular, the expression of NAE1 was significantly downregulated in ovarian cancer samples (Figure 2A). In addition, in three independent Gene Expression Omnibus (GEO) datasets (GSE14407, GSE26712, and GSE40595), NAE1 was downregulated in ovarian tumors compared with normal ovarian surface epithelium (Figure 2B). To determine the relationship between the transcription level of NAE1 and the pathological stage of ovarian cancer, we used the GEPIA stage plot. The mRNA expression of NAE1 was negatively related to the tumor stage of ovarian cancer (Figure 2C). Moreover, using GEO datasets (GSE17308), we confirmed that the expression of NAE1 was slightly decreased in benign tumors and further decreased in invasive cancers compared with normal tissues (Figure 2D). The tumor suppressor gene *TP53* is disrupted in more than half of human cancers [15], among which ovarian cancer has the highest mutation frequency at approximately 80% (Figure 2E). Most of these are known as missense or null mutations, and null mutation of *TP53* accounts for about 38% in early stage ovarian cancer [32]. Moreover, approximately 96% of high-grade serous ovarian cancer (HGSOC), the most common and aggressive subtype of ovarian cancer, exhibits *TP53* deletion and mutation [33], which implies that p53 disruption and reduced NAE1 expression may be associated with cancer progression. In addition, ovarian cancer cells showed dramatically different outcomes according to p53 status. Therefore, our subsequent mechanistic studies were performed in ovarian cancer cell lines.

### 3.3. Slug Plays a Major Role in Regulating Neddylation Blockade-Mediated EMT in Cancer Cells

To identify the molecular mechanisms underlying neddylation blockade-mediated migration, we compared the expression levels of various EMT-related markers between wild type p53 (PA-1, A2780) and p53-null (SKOV-3) ovarian cancer cell lines by Western blotting. Among various markers, the EMT-promoting transcription factor Slug was notably reduced in PA-1 and A2780 cells, but elevated in SKOV-3 cells, following MLN4924 treatment (Figure 3A). This was further confirmed in cells transfected with si-NEDD8 RNA (Figure 3B). Enhanced Slug expression was also observed in other p53-null cancer cell lines, but cells expressing wild type p53 did not show any change in Slug expression by neddylation blockade (Appendix A). Furthermore, we observed enhanced expression of other EMT activator (Zeb1 and Snail) by neddylation blockade in SKOV-3 cells. Consistent with previous studies regarding that neddylation blockade promotes EMT through induction of hypoxia-inducible factor 1α (HIF-1α) [25] and EMT-promoter Zeb1 and Snail are directly regulated by HIF-1α [34,35], these factors might be possible regulator of EMT in SKOV-3. Nevertheless, there was no p53 status-dependent difference in these factors, so we focused on the mechanism of Slug.

Because Slug induces cancer EMT by transcriptional inhibition of E-cadherin, a major epithelial marker, we confirmed that E-cadherin expression was opposite to that of Slug (Figure 3A,B). We then conducted immunofluorescence analysis and showed that E-cadherin expression was augmented in the presence of p53 but reduced in p53-null cell lines upon neddylation blockade (Figure 3C). Next, we examined whether the expression levels of NAE1 and CDH1 (E-cadherin) were correlated in ovarian cancers, which have high disruption of p53. A dataset from patients with ovarian cancer showed a strong correlation between NAE1 and CDH1 (r = 0.5, *p* = 1 × 10^−27^) (Figure 3D). To further validate these findings, we silenced Slug in SKOV-3 cells and demonstrated that knockdown of Slug inhibited neddylation blockade-induced cell migration (Figure 3E,F). This result was confirmed in other p53-null cancer cell lines (Appendix A). Overall, these data indicated that Slug mediates the EMT process induced by neddylation blockade in cancer cells.

### 3.4. Neddylation Blockade Promotes Proteasomal Degradation of Slug in Cancer Cells Expressing Wild Type p53

Based on the above results, we evaluated whether p53 is responsible for the reduction in Slug expression in cancer cells with wild type p53 expression. In cancer cell lines expressing wild type p53, treatment with MLN4924 attenuated Slug expression in dose- and time-dependent manners (Figure 4A). To explore this issue, we used proteasome inhibitor MG132 and found that Slug undergoes proteasomal degradation when neddylation was inhibited (Figure 4B). In addition, it was confirmed that Slug induction by MLN4924 in SKOV-3 was suppressed by wild type p53 expression, which was recovered with MG132 treatment (Figure 4C). These results demonstrate that neddylation blockade promotes proteasomal degradation of Slug in cancer cells expressing wild type p53.

Previous studies showed that p53 and its downstream targets, p21 and MDM2, bound to Slug and induced its ubiquitin-mediated proteasomal degradation [12,13]. To verify whether this occurs in ovarian cancer cells, we detected interactions among the aforementioned factors by immunoprecipitation (Figure 4D). We then performed a loss-of-function study using two different siRNA sequences targeting p53, which demonstrated that p53 knockdown led to elevated expression of Slug. In the same context, overexpression of wild type p53 reduced Slug protein levels in SKOV-3 cells in a dose-dependent manner (Figure 4E). To further confirm that these results occurred at post-translation level, we confirmed that there was no change in Slug mRNA for p53-knockdown in PA-1 cells and p53-overexpression in SKOV-3 cells (Figure 4F and Appendix A). Similarly, we showed that p53 expression reduced Slug protein stability by performing cycloheximide chase assay (Figure 4G).

Given that NEDD8 conjugation to p53 reduces its transcriptional activity [18], we performed luciferase reporter assays to evaluate how neddylation blockade affects p53 function. Both endogenous and exogenous p53 showed enhanced luciferase activity after neddylation was blocked (Figure 4H). Consistent with this finding, both the mRNA and protein levels of MDM2 and p21 were elevated following neddylation blockade (Figure 4I,J). Our results indicated that Slug ubiquitination was enhanced by neddylation blockade but fully abolished by p53 knockdown (Figure 4K). Thus, our findings demonstrate that neddylation blockade promotes expression of MDM2 and p21 by increasing p53 transcriptional activity, thereby enhancing the level of the p53/p21/MDM2/Slug complex leading to Slug protein degradation.

### 3.5. Interaction of Mutant p53 with Slug Neutralizes the Effect of Neddylation Blockade on Cell Migration

p53 is mutated mainly in its DNA-binding domain, which interferes with the ability of p53 to bind to the promoter regions of target genes [36]. To explore the effect of neddylation blockade on the migration of mutant p53-expressing cancer cells, we first examined the expression levels of Slug in three p53-mutant cancer cell lines. Our results showed no significant changes in Slug protein level between vehicle-treated and MLN4924-treated cells (Figure 5A). Furthermore, transfection of ES-2 cells with si-NEDD8 did not affect the Slug expression level (Figure 5B). Next, we performed wound healing and Transwell assays using MLN4924 and si-NEDD8, which demonstrated that neddylation blockade did not affect ES-2 cell migration (Figure 5C,D). For additional confirmation, we evaluated the transcriptional activity of p53 and mRNA levels of its downstream targets MDM2 and p21. As expected, neddylation blockade did not increase transcriptional activity of p53 (Figure 5E,F). Furthermore, we performed immunoprecipitation to assess whether mutant p53 binds to Slug in ES-2 cells and observed interactions among p53, Slug, p21, and MDM2 (Figure 5G). We then silenced mutant p53 protein and evaluated Slug expression and cell migration mediated by neddylation blockade. Mutant p53 knockdown induced Slug expression and cell migration of ES-2 cells. Moreover, neddylation blockade further enhanced Slug expression (intensity ratio of Western blot band, 1: 2.39) and slight cell migration (*p* value = 0.55) in mutant p53-silenced ES-2 cells (Figure 5H). Our results indicated that mutant p53 protein maintained binding to Slug and suppressed cell migration mediated by neddylation blockade. However, mutant p53, unlike wild type p53, did not promote further degradation of Slug associated with neddylation blockade because this form of p53 was transcriptionally inactivated.

### 3.6. Neddylation Blockade Enhances Slug Expression via the PI3K/Akt/mTOR/Slug Axis in p53-Null Cells

Consistent with the above results showing that neddylation blockade enhanced cell migration and Slug expression in p53-null cancer cell lines, MLN4924 increased the Slug protein level in SKOV-3 cells in dose- and time-dependent manners (Figure 6A). RT-qPCR analysis indicated that the increased protein level was attributed to increased transcription (Figure 6B). Because Slug is regulated by various signaling pathways [37,38,39], we screened well-known upstream mediators by Western blot assays. In line with our previous study suggesting that neddylation blockade facilitated p53-null cell migration via the PI3K/Akt/mTOR pathway [25], p-Akt was activated during neddylation blockade (Figure 6C). We then investigated how neddylation blockade affects cells with different states of p53, and found that active Akt was induced by neddylation blockade in mutant p53-expressing ES-2, but not in cancer cells expressing wild type p53 (Appendix A). In addition, neddylation blockade enhanced Slug mRNA expression in p53-mutant cells, but did not in wild type p53 cell lines (Appendix A). Induction of Slug mRNA levels by neddylation blockade was observed in wild type p53 expressing SKOV-3, but the degree of increase was relatively less than that without p53 (Appendix A and Figure 6B). These results may be consistent with the previous studies showing that p53 increases the expression of genes (e.g., PTEN and AMPK) which inhibit Akt/mTOR signaling pathway [40].

It was reported that the activation of the PI3K/Akt/mTOR pathway increased Slug expression in lung adenocarcinoma cells [41]. Therefore, we confirmed that Slug expression was regulated by the PI3K/Akt/mTOR signaling axis during MLN4924 treatment by applying inhibitors of PI3K (LY294002), Akt (MK-2206), and mTOR (rapamycin) (Figure 6D,E). Furthermore, we assessed SKOV-3 cell migration by wound healing and Transwell assays, which demonstrated that the neddylation blockade-associated enhanced cell migration was remarkably suppressed by the above inhibitors (Figure 6F,G). Therefore, our results implied that neddylation blockade enhances Slug expression via the PI3K/Akt/mTOR/Slug signaling axis in p53-null cancer cell lines.

### 3.7. p53 Presence Determines Neddylation Blockade-Mediated Cancer Cell Migration

Last, we assessed whether the previous findings were maintained in a polydimethylsiloxane (PDMS) three-dimensional spheroid culture system that mimicked in vivo tumor conditions (Figure 7A) [42]. Thus, we cultured cells in PDMS chips and assessed cell migration by measuring spheroid roundness on day 5, as described previously (Appendix A) [43]. We found that the anti-migration effect of neddylation blockade was absent after p53 knockdown in cancer cells expressing wild type p53. In addition, MLN4924-induced SKOV-3 cell migration was inhibited after ectopic expression of wild type p53 (Figure 7B). We then confirmed that the reduced Slug protein expression due to neddylation blockade was recovered after p53 knockdown in wild type p53 cell lines. Furthermore, transient transfection of wild-type p53 in SKOV-3 cells resulted in the repression of Slug expression by neddylation blockade (Figure 7C). Taken together, our results indicate that the p53 expression status in cancer cells determines neddylation blockade-mediated cancer cell migration.

## 4. Discussion

Although neddylation is an important factor in cancer metastasis [20,28], the cause of paradoxical neddylation results among cancer cells has not been clearly elucidated. In this study, we found that the expression status of the p53 tumor suppressor in cancer cells determined the fate of neddylation blockade-mediated cell migration. The regulatory domain of p53 is reportedly neddylated by MDM2, thereby inhibiting its transcriptional activity [18]. Another well-known substrate of neddylation is the Cullin subunit of Cullin-RING ligases, a large family of E3 ubiquitin ligases. Cullin-RING ligases can be activated by neddylation and contribute to the ubiquitination of cellular proteins involved in cell cycle arrest and apoptosis [44]. Accordingly, overactivated neddylation and high expression of the NEDD8-activating enzyme 1 (NAE1) have been found in human cancers such as hepatocellular carcinoma, osteosarcoma, glioblastoma, lung cancer, and colorectal cancer [22,23,30,31,45]. The first-in-class NAE1 inhibitor, MLN4924, has therapeutic effects on various cancers and is the focus of clinical trials for potential use as an anticancer agent [46]. In contrast to the previous hypotheses, neddylation blockade has recently been shown to induce cancer cell migration in various cancer cell lines [20,24,25,28]. For example, Zhou et al. found that disruption of neddylation promoted in vivo tumor formation and cell migration in cancer stem cells [24]. Moreover, our previous study showed that neddylation blockade enhanced cancer cell migration via activation of the PI3K/Akt pathway by inhibiting C-CBL-mediated c-Src neddylation and poly-ubiquitination [20]. Surprisingly, we found that the cancer cell lines used in our studies (SKOV-3, PC3, and H1299 cells) all exhibit p53 disruption [25]. Moreover, based on the observations that NAE1 expression is downregulated in malignant ovarian tumors and *TP53* is highly disrupted in ovarian cancer samples (Figure 2), we hypothesized that the p53 status may determine neddylation-induced cancer cell migration.

Because almost half of all human cancers are not detected until the late stages (III or IV), in which cancers are already disseminated to other sites in the body, the inhibition of migration is an important therapeutic target for human malignancies [47]. Various genetic alterations are associated with changes in tumor stage, among which *TP53* is the most representative driver gene [48]. p53 plays a role in antitumor activities by transcriptionally regulating genes involved in DNA repair, cell cycle arrest, apoptosis, and other processes [6]. Importantly, p53 inhibits cancer EMT [5,49], an important process that confers invasive properties to epithelial cells by reducing cell–cell adhesion and polarity [7]. Because cancer cells metastasize via the acquisition of mesenchymal properties, EMT represents a major prerequisite for tumor progression [50]. Cancer EMT is induced by EMT-promoting transcription factors such as Slug, Snail, ZEB1, and Twist. These factors repress transcription of epithelial-related gene E-cadherin by binding to E-box domain of its promoter region [9]. Because these EMT-activators are controlled by p53, we screened for changes in EMT-related proteins following neddylation blockade and found that Slug induced cancer cell migration according to p53 status (Figure 3).

Slug is overexpressed in numerous malignant cancers, including leukemia, lung, liver, breast, esophageal, colorectal, gastric, pancreatic, prostate, and ovarian cancers [51]. Previous studies showed that Slug plays a vital role in the development of cancer progression by stimulating invasiveness and tumor metastasis [52,53]. For example, the overexpression of Slug promoted cancer cell migration, and knockdown of Slug reduced in vivo tumor metastasis [52,54]. Thus, the modulation of Slug expression is critical for EMT and cancer progression. Slug can be controlled by diverse post-translational modifications. For instance, Glycogen synthase kinase 3 beta (GSK3β)-mediated phosphorylation of Slug triggers its degradation [55]. In addition, p53, p21, and MDM2 interact with Slug, leading to its ubiquitination [12,13]. Considering that neddylation reduces the transcriptional activity of p53 [18], and that p21 and MDM2 are representative downstream genes [56], we presumed that neddylation blockade would increase p21 and MDM2 expression by inducing p53 transcriptional activity. Increased levels of the p53/p21/MDM2/Slug complex destroyed Slug protein by promoting its ubiquitination in cancer cell lines expressing wild type p53. Here, we confirmed that neddylation blockade led to enhanced Slug ubiquitination, and this effect was negated by knockdown of p53 (Figure 4K). Moreover, Slug is reportedly targeted by a complex network of oncogenic signaling pathways such as Transforming growth factor-β (TGF-β)/Smad, Signal transducer and activator of transcription 3 (STAT3), and the PI3K/Akt/mTOR signaling pathway [37,39,41,57,58]. Consistent with our previous findings that neddylation blockade stimulated the PI3K/Akt/mTOR pathway in various p53-null cancer cell lines [20,25], we found that Slug transcription is primarily regulated by this signaling axis in p53-null cell line (Figure 6). In addition, the enhanced expressions of p-Akt and Slug by neddylation blockade were observed in mutant p53-expressing ES-2 cells, but not in wild-type p53 cell lines (Appendix A). These results may correspond to previous report that the induction of p53 target genes such as PTEN, tuberous sclerosis protein 2, beta subunit of AMP-activated protein kinase, and IGF-binding protein-3 inhibit Akt/mTOR signaling pathway [40]. Recent evidence indicates that mutations in p53 not only causes loss of function, but also leads to gain of function of p53. Notably, p53 mutants acquire oncogenic characteristics and promote tumor aggressiveness [59]. As gain of function of mutant p53 facilitates EMT and cancer metastasis [60], we investigated whether neddylation blockade affects the activity of mutant p53 and the expression of Slug. We showed that neddylation blockade did not influence the Slug protein level or cell migration in p53-mutat cell lines. In addition, we discovered that mutant p53 protein maintained its ability to bind to Slug (Figure 5). Meanwhile, our previous studies using different p53 mutant-expressing cancer cell lines (U373MG and U251MG) showed increased cell migration upon blocking neddylation [20,28]. p53 interacts with Slug via its DNA-binding domain, and 97% of p53 mutations occur in the DNA-binding domain [12,36]. Therefore, we assumed that the interactions of p53 with MDM2, p21, and Slug depend on the mutation sites within the DNA-binding domain that lead to conformational changes in p53, thereby enabling formation of complexes between MDM2 or p21 with Slug. Thus, disruption of p53 in combination with downregulation of NAE1 could be potential prognostic factors that accelerate cancer metastasis (Figure 2). Collectively, we demonstrated a p53-dependent mechanism of neddylation that regulates cancer cell migration, particularly via regulation of Slug expression. Further research is needed to investigate differences in cancer cell migration following site-specific mutations in p53 that influence neddylation.

## 5. Conclusions

In summary (Figure 8), we demonstrated that the p53 expression status affects Slug expression, which is a major determinant of cell migration induced by neddylation blockade. Indeed, multiple studies have identified MLN4924 as a potential anticancer agent in various cancer types. Nonetheless, our data imply that careful attention is needed concerning the development of neddylation blockade as a therapeutic target based on the cancer genetic background.

## Figures and Tables

**Figure 1 cancers-13-00531-f001:**
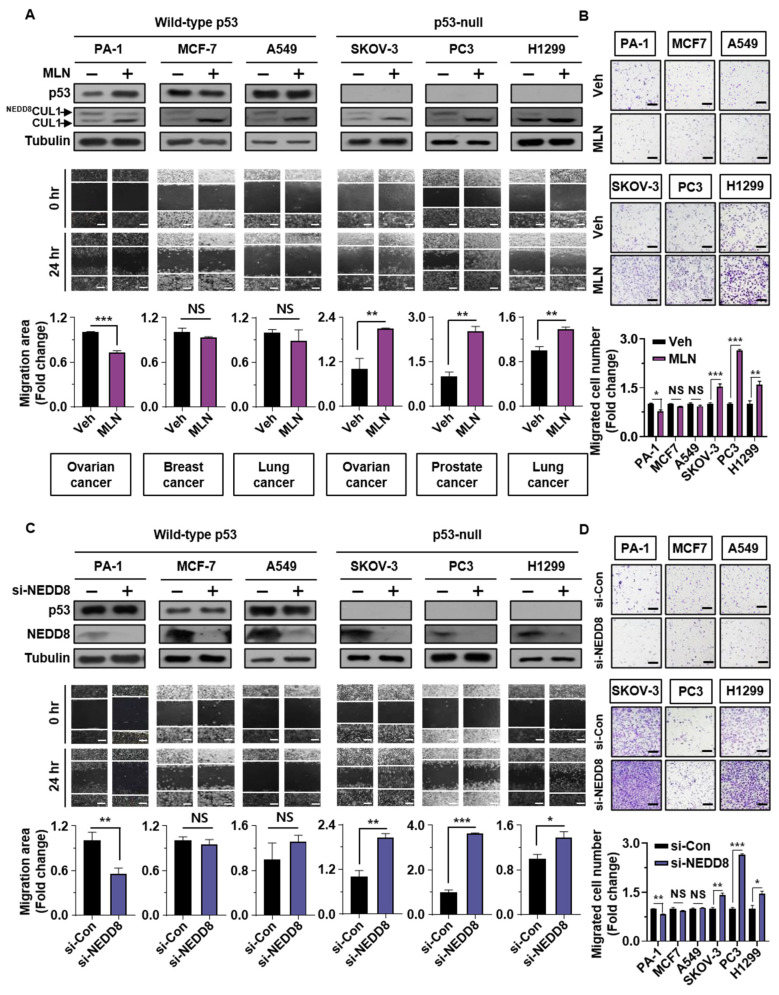
Neddylation blockade induces cancer cell migration according to p53 status. (**A**,**C**) Wild type p53 (PA-1, MCF-7, and A549) and p53-null (SKOV-3, PC3, and H1299) cancer cell lines were subjected to wound healing assays with or without MLN4924 (125 nM) for 24 h or were transfected with si-Control or si-NEDD8 and analyzed by Western blotting. Scale bar: 200 μm. Whole areas were measured using ImageJ software. The upper band of CUL1 indicates neddylated CUL1, and the lower band indicates CUL1. (**B**,**D**) Cancer cells treated with or without MLN4924 (125 nM) for 24 h or transfected with si-Control or si-NEDD8 were subjected to Transwell assays. The number of cells in four randomly chosen fields was counted. Data are presented as the means ± standard deviation (*n* = 3). * *p* < 0.05; ** *p* < 0.01; *** *p* < 0.001; NS, not significant.

**Figure 2 cancers-13-00531-f002:**
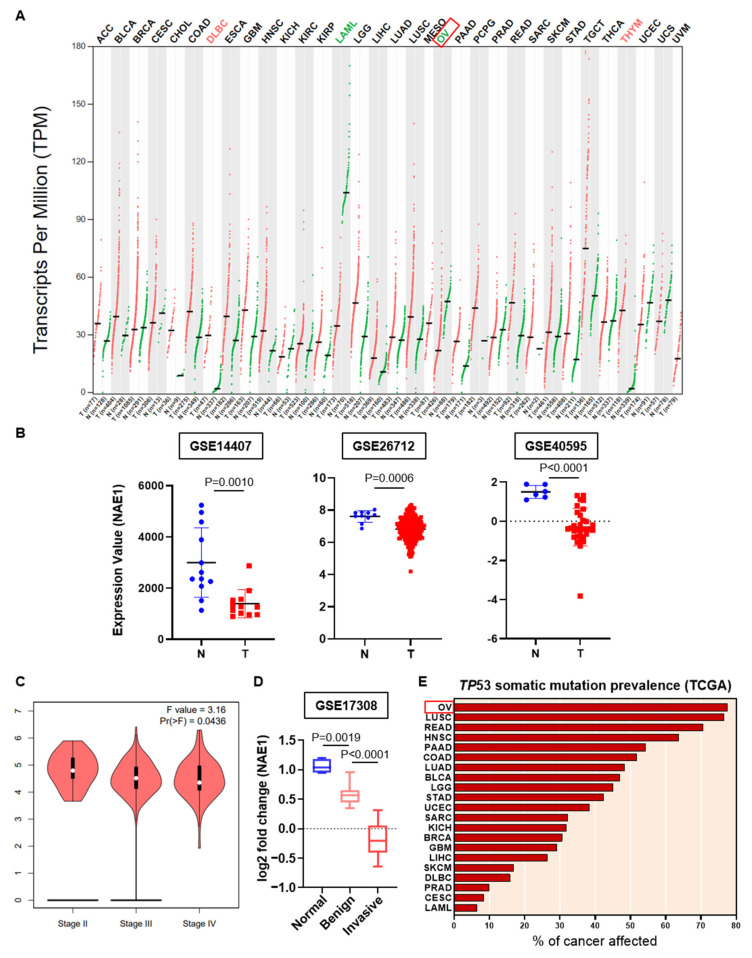
Associations of NAE1 expression, the pathological stage, and *TP53* mutation prevalence in ovarian tumors. (**A**) Expression level (transcript per million: log2(TPM + 1)) of NAE1 between TCGA tumors (red dots) and TCGA normal, and GTEx normal data (green dots) in 27 cancer types using Gene Expression Profiling Interactive Analysis (GEPIA). T: Tumor; N: Normal; n: number. (**B**) Distribution of NAE1 expression in normal ovarian surface epithelium (blue circles) and serous ovarian cancer epithelium (red squares) (GSE14407, GSE26712, GSE40595). (**C**) Violin plot indicating the correlation between NAE1 expression and pathological stage in patients with ovarian cancer (GEPIA). The *p*-value was set at 0.05. (**D**) Log2 fold change of NAE1 mRNA expression in ovarian surface epithelium (blue box), benign ovarian serous tumors (pink box), and invasive ovarian serous tumors (red box) (GSE17308). (**E**) *TP53* somatic mutation prevalence in TCGA data were downloaded from the International Cancer Genome Consortium (ICGC) web portal.

**Figure 3 cancers-13-00531-f003:**
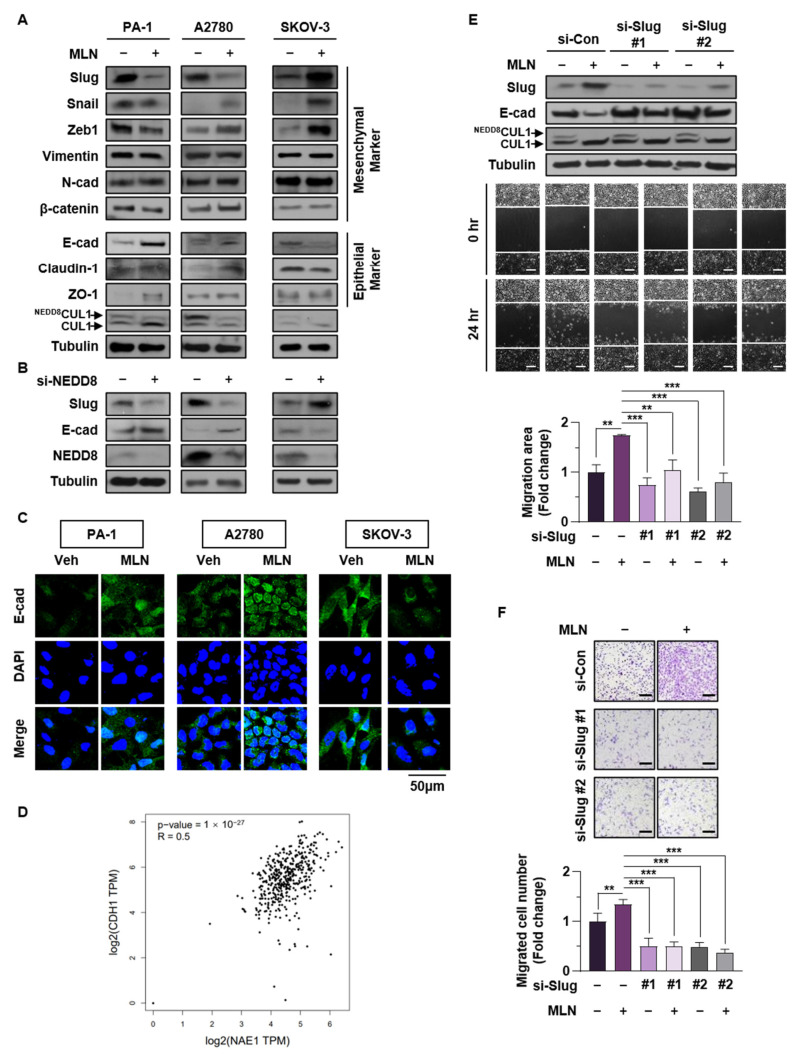
Slug plays a major role in regulating neddylation blockade-mediated EMT in cancer cells (**A**,**B**) PA-1, A2780, and SKOV-3 cells were treated with or without MLN4924 (125 nM) or transfected with si-Control or si-NEDD8. Protein levels of representative EMT markers were verified by western blot assays. (**C**) After cells had been treated with or without MLN4924 for 24 h in serum free medium, the expression of E-cadherin (green) was visualized by immunofluorescence. Scale bar: 50 μm. The results shown are representative of three experiments with similar results. (**D**) Correlation between expression of NAE1 and CDH1 in ovarian cancer, analyzed using the GEPIA database. (**E**) SKOV-3 cells transfected with si-Control or si-Slug#1, 2 and/or treated with MLN4924 (125 nM) were subjected to wound healing assays. Cell lysates were then subjected to western blot assays to verify protein levels of Slug and E-cadherin. Scale bar: 200 μm. Whole areas were measured using ImageJ software, and data are presented as the means ± standard deviation (*n* = 3). (**F**) SKOV-3 cells transfected with si-Control or si-Slug#1, 2 and/or treated with MLN4924 (125 nM) were subjected to Transwell assay. The number of cells in four randomly chosen fields was counted. ** *p* < 0.01; *** *p* < 0.001; NS, not significant.

**Figure 4 cancers-13-00531-f004:**
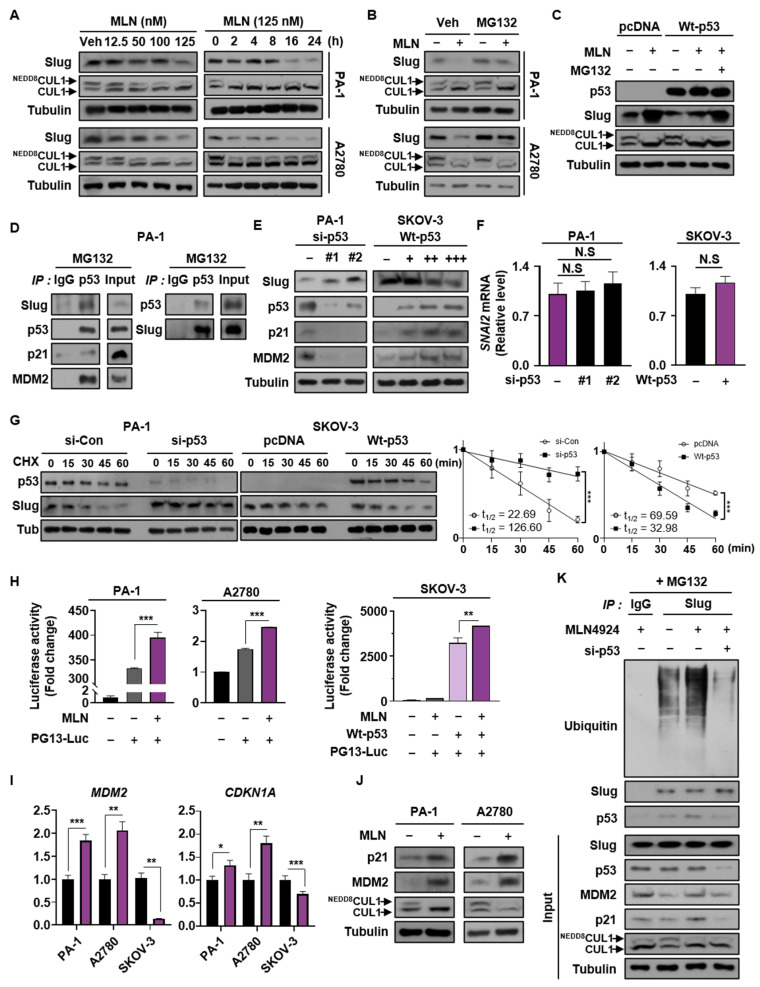
Neddylation blockade promotes proteasomal degradation of Slug in cancer cells expressing wild type p53. (**A**) PA-1 and A2780 cells were treated with MLN4924 at different doses and times. Slug expression was verified by Western blot analysis. (**B**) Cells treated with or without MLN4924 for 24 h were incubated with or without MG132 for 6 h, and the cell lysates were subjected to Western blot assays. (**C**) SKOV-3 cells transfected with pcDNA or Wt-p53 were incubated with or without MLN4924 and MG132, and the cell lysates were subjected to Western blot assays (**D**) PA-1 cells were treated with MG132 for 6 h, and the cell lysates were assessed by immunoprecipitation using anti-p53 or anti-Slug antibodies. Precipitated proteins were analyzed by Western blot assays. (**E**) PA-1 cells transfected with si-Control or si-p53 RNAs (#1, #2) and SKOV-3 cells transfected with different doses of wild type p53 DNA were subjected to immunoblotting. (**F**) PA-1 cells transfected with si-Control, si-p53#1, and si-p53#2, and SKOV-3 cells transfected with pcDNA and wild type p53 were subjected to RT-qPCR. (**G**) si-Control or si-p53 transfected PA-1 cells and pcDNA or wild type p53 transfected SKOV-3 cells were treated with cycloheximide for the indicated time, and then cell lysates were subjected to Western blotting. Band intensities (mean ± SD, *n* = 3) on blots were analyzed using ImageJ and plotted. (**H**) Cells were co-transfected with luciferase plasmid and the other plasmid shown and then subjected to luciferase reporter assays. The results were normalized to β-galactosidase activity. (**I**,**J**) Cells were treated with or without MLN4924 and subjected to RT-qPCR and Western blot assays to analyze mRNA and protein levels of MDM2 and p21. Data are expressed as means ± standard deviation (*n* = 3). * *p* < 0.05; ** *p* < 0; *** *p* < 0.001; NS, not significant. (**K**) A2780 cells transfected with si-Control or si-p53 #1 were treated with or without MLN4924 for 24 h and then with MG132 for 6 h. Cell lysates were then precipitated using an anti-Slug antibody, and the precipitates were analyzed by western blot assays.

**Figure 5 cancers-13-00531-f005:**
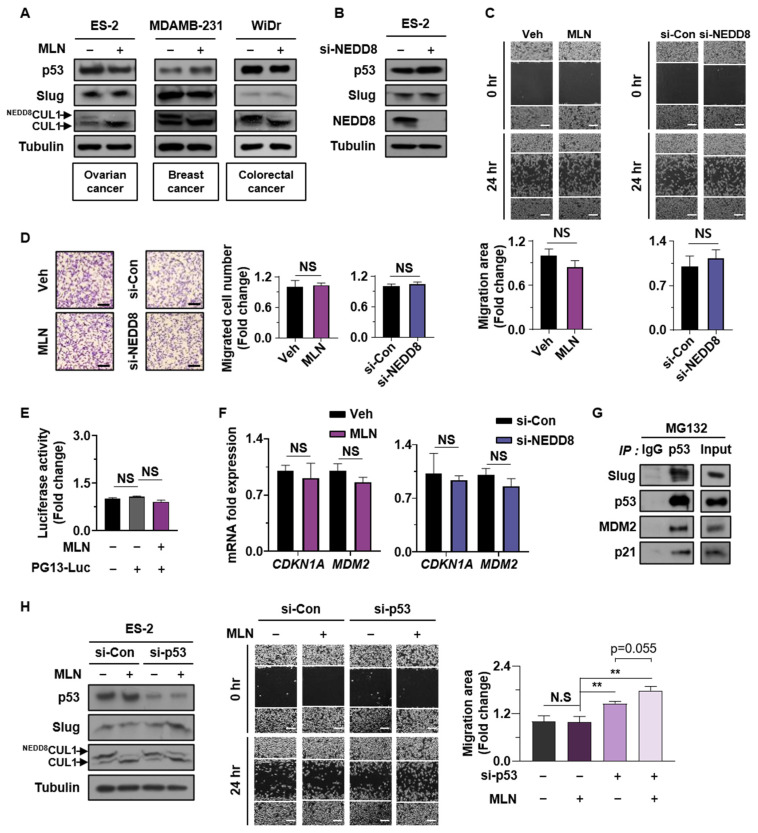
Interaction of mutant p53 with Slug neutralizes the effect of neddylation on cell migration. (**A**,**B**) ES-2, MDAMB-231, and WiDr cells treated with vehicle or MLN4924, and ES-2 cells transfected with si-Control or si-NEDD8 were subjected to Western blot assays. (**C**,**D**) ES-2 cells were subjected to wound healing and Transwell assays with or without MLN4924 or transfected with si-Control or si-NEDD8. Scale bar: 200 μm. Whole areas were measured using ImageJ software. The numbers of cells in four randomly chosen fields were counted. Data are presented as the means ± standard deviation (*n* = 3). (**E**) ES-2 cells were co-transfected with luciferase plasmid and the other plasmid shown and then subjected to luciferase reporter assays. The results were normalized to β-galactosidase activity. Data are expressed as means ± standard deviation (*n* = 3). (**F**) ES-2 cells were treated with or without MLN4924 for 16 h and subjected to RT-qPCR to analyze mRNA levels of MDM2 and p21. Data are expressed as means ± standard deviation (*n* = 3). (**G**) ES-2 cells treated with MG132 for 6 h, and the cell lysates were assessed by immunoprecipitation using an anti-p53 antibody. Precipitated proteins were analyzed by Western blot assays. (**H**) ES-2 cells transfected with si-Control or si-p53 and treated with or without MLN4924 were subjected to wound healing assay and western blot. Scale bar: 200 μm. Whole areas were measured using ImageJ software. The numbers of cells in four randomly chosen fields were counted. Data are presented as the means ± standard deviation (*n* = 3). ** *p* < 0.01; NS, not significant.

**Figure 6 cancers-13-00531-f006:**
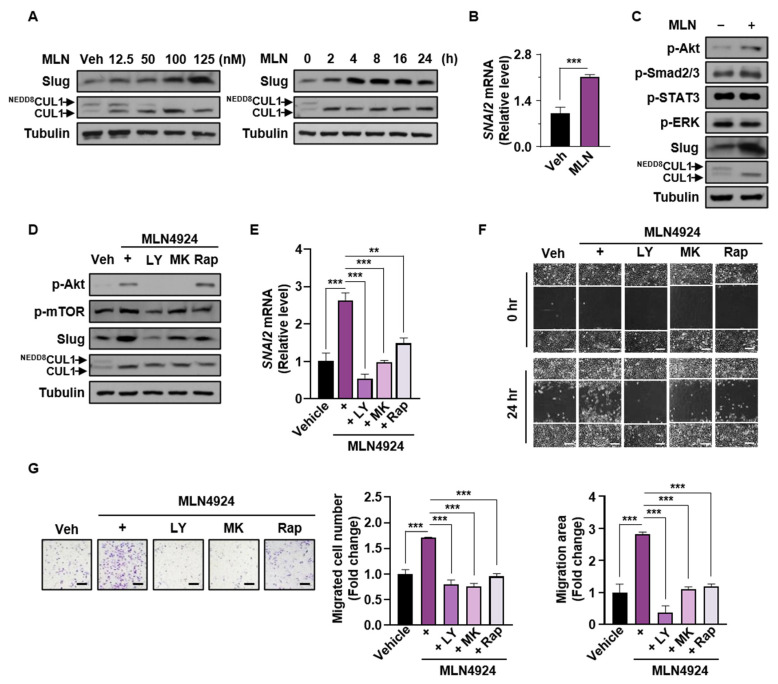
Neddylation blockade enhances expression of Slug via the PI3K/Akt/mTOR/Slug axis in p53-null cancer cells. (**A**) SKOV-3 cells were treated with MLN4924 at different doses and times and then subjected to Western blot analysis. (**B**) SKOV-3 cells treated with or without MLN4924 were subjected to RT-qPCR. Data are expressed as means ± standard deviation (*n* = 3). ** *p* < 0.01; *** *p* < 0.001; NS, not significant. (**C**) SKOV-3 cells were treated with vehicle or MLN4924, and the proteins shown were analyzed by Western blot assays. (**D**,**E**) SKOV-3 cells were treated with MLN4924 and the indicated inhibitors for and then subjected to Western blot and RT-qPCR assays. Data are expressed as means ± standard deviation (*n* = 3). ** *p* < 0.01; *** *p* < 0.001; NS, not significant. (**F**) SKOV-3 cells were treated with MLN4924 and the indicated inhibitors and then subjected to wound healing assays. Scale bar: 200 μm. Whole areas were measured using ImageJ software. (**G**) SKOV-3 cells incubated with MLN4924 and the indicated inhibitors were subjected to Transwell assays. Scale bar: 200 μm. Data are presented as the means ± standard deviation (*n* = 3). ** *p* < 0.01; *** *p* < 0.001; NS, not significant.

**Figure 7 cancers-13-00531-f007:**
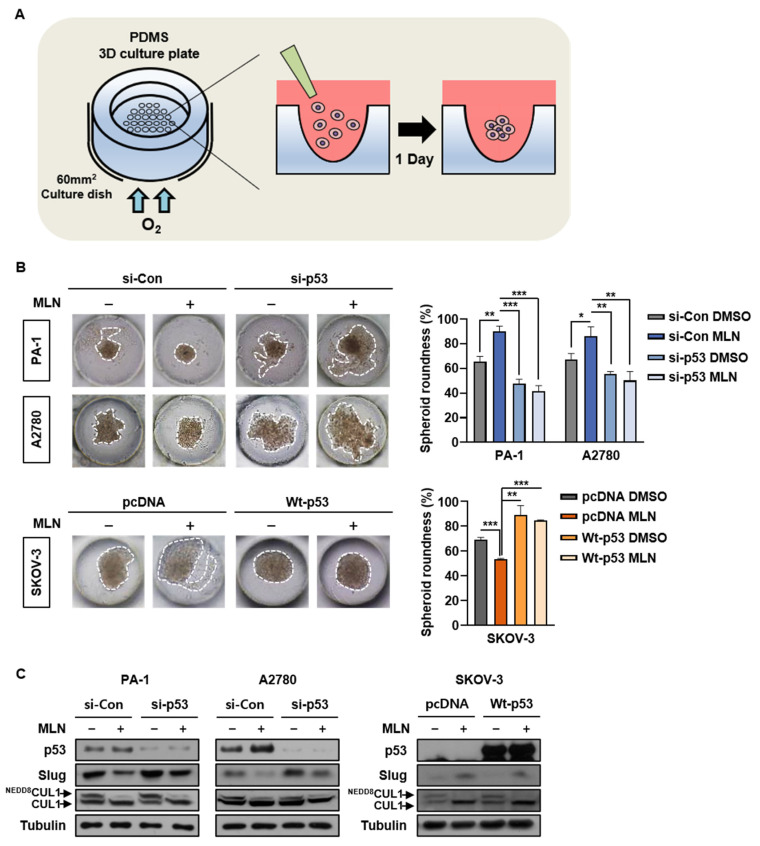
Presence of p53 determines neddylation blockade-mediated cancer cell migration. (**A**) Schematic of polydimethylsiloxane three-dimensional spheroid cultures. (**B**,**C**) PA-1 and A2780 cells transfected with si-Control or si-p53 and SKOV-3 cells transfected with pcDNA or wild type p53 were incubated in PDMS 3D culture chips with culture medium containing vehicle or MLN4924. Cell lysates were then subjected to western blotting. Representative optical microscopy images were obtained on day 5. The average spheroid diameter was measured using ImageJ, and data are presented as the means ± standard deviation (*n* = 3). * *p* < 0.05; ** *p* < 0.01; *** *p* < 0.001; NS, not significant.

**Figure 8 cancers-13-00531-f008:**
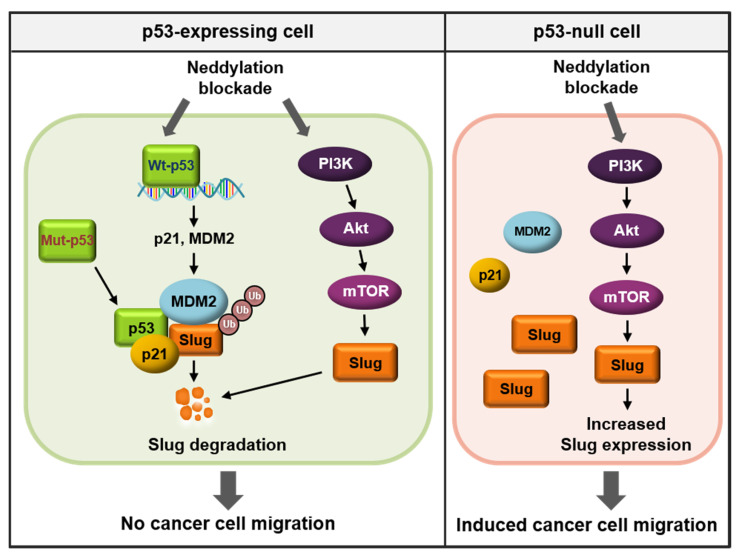
Schematic illustrating the different roles of neddylation blockade according to p53 status.

## Data Availability

Data available in a publicly accessible repository that does not issue DOIs. Publicly available datasets were analyzed in this study. This data can be found here: [https://www.ncbi.nlm.nih.gov/geo/query/acc.cgi?acc=GSE14407/accession number: GSE14407], [https://www.ncbi.nlm.nih.gov/geo/query/acc.cgi?acc=GSE26712/accession number: GSE26712], [https://www.ncbi.nlm.nih.gov/geo/query/acc.cgi?acc=GSE40595/accession number: GSE40595], [https://www.ncbi.nlm.nih.gov/geo/query/acc.cgi?acc=GSE17308/accession number: GSE17308].

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
