# Peer review of "The Effect of Neddylation Blockade on Slug-Dependent Cancer Cell Migration Is Regulated by p53 Mutation Status"

_cancers, 2021, doi:10.3390/cancers13030531_

Round 1
Reviewer 1 Report
This is an interesting and well-designed study that aims to resolve the controversy surrounding the opposing effects of neddylation blockade on cell migration in various cancer types. The authors report that the effect of neddylation blockade on cell migration depends on the loss of p53 transcriptional activity; the neddylation inhibitor MLN4924 induces cell migration in various types of cancer cells that are p53 null or mutant but has no such effect on p53 wild-type cells. Furthermore, they show that MLN4924 induces cell migration by activating the EMT-promoting transcription factor Slug via the PI3K/Akt/mTOR pathway, whereas Slug is degraded in p53 wild-type cells via the p53/p21/MDM2 pathway. Therefore, the study has therapeutic implications with regards to precision targeting in cancer treatment.
The results are well controlled and presented logically and overall support the conclusions. Some specific comments listed below need to be addressed to improve clarity.
Specific comments:
1. The title of the manuscript “…Regulated by p53 Expression Status” is confusing; it is p53 mutation rather than the expression level that is most relevant here.
2. Line 62: the statement “These factors promote EMT by binding to E-box domain of E-cadherin” is unclear and can be construed as protein-protein interactions between EMT-related transcription factors and E-cadherin.
3. Nonstandard abbreviations, such as N8, used in figure legends need to be defined. It is also unclear what immunoblotting of CUL1 (e.g., in Fig. 1A) is for and what the double and single bands mean.
4. Although Fig. 3 supports an important role of Slug in MLN4924-induced EMT in p53-null cells, the possible involvement Snail and Zeb1 (also upregulated) was neither examined nor discussed, particularly in consideration of the authors’ recent publication (10.1038/s41598-020-75286-0).
Reviewer 2 Report
In the manuscript titled “The effect if Neddylation Blockade on Slug-dependent cancer cell migration is regulated by p53 expression statis” Kim Y et al., seek to determine the effect of p53 mutation status of cancer cells on neddylation-mediated cell migration. They show that neddylation blockade induced cell migration is dependent on the p53 status of the cancer cells and mediated via the transcription factor Slug. Neddylation blockade induced proteasome-mediated Slug degradation in p53-WT cell. In cells that lack p53, neddylation blockade increased cell migration. Neddylation blockade did not impact the migration of cells that express p53 mutants. They also provide experimental evidence suggesting the role of Slug in mediating this response.
Overall, the study presents findings that will improve our understanding of mechanisms by which cancer cell migration can be impacted by neddylation in a p53-depdendent manner. Given, the widespread prevalence of p53 mutations in human cancer this will be of interest to the community. However, there are certain issues that will need to be addressed before publications. I recommend the paper for publication once the following issues (listed below) have been resolved:
Major:
- If neddylation blockade causes Slug degradation and depletion in p53-WT cells, then why don’t we see decreased cell migration in p53-WT (MCF-7 and A-549) cells in Fig 1A? Have the authors checked what happens to Slug in these cells upon neddylation blockade? Is Slug depletion in these cells counteracted by another mechanism?
- Fig 4E shows that sip53 in p53 WT cells causes increased Slug protein levels. Do you see an increase in slug RNA level or is the effect post-translational? Also, ectopic WT-p53 expression in SKOV3 cells causes decrease in Slug protein level. Is this effect, transcriptional or post-translational? Authors need to perform these two experiments to address these questions clearly:
- qRT PCR to determine if there is a change in SNAI2 mRNA in response to p53 knockdown in PA-1 cells and overexpression in SKOV3 cells.
- Cycloheximide chase, to ascertain the effect is post-translational and mediated at the protein stability level under the same conditions listed above.
- In Fig.6 the authors show that the effect of MLN is mediated via transcriptional upregulation of SLUG in p53-null SKOV3 cells. What happens if you do the same experiment in SKOV3 cells that expresses p53-WT or mutant protein? Is Slug still transcriptionally upregulated?
- Does p53-WT or mutant p53 expression in SKOV3 abolish MLN and si-N8 differences observed in cell migration? Or if your knock down p53-S241F in ES-2 cells does it make it responsive to the above treatments?
- Why is there is a discrepancy between Fig 5A and Fig 5B. In 5A, MLN (neddylation inhibitor) caused no change in Slug expression in ES-2 cells as per the Western blot. However, in Fig 5B in ES-2 cells si-N8 resulted in decreased Slug protein level as per the western blot.
Minor:
- No data provided indicates the claim made in lines 366-368. The statement needs to be revised or supportive data has to be provided.
- In Fig 5B Si-N8 resulted in decreased Slug protein levels in ES-2 cells. However, line 394-395 states that Si-N8 did not affect Slug expression level.
Reviewer 3 Report
In this manuscript, the authors have tried to understand whether p53 status of cancer influence neddylation blockade induced migration of cancer cells.
Major Comments:
- The authors should look at the expression of other NEDD8 pathway markers such as APP-BP1, Ubc 12 or even NEDD8 expression after blocking with MLN4924.
- The authors should introduce wild type or mutant p53 in all three p53 null cell lines to show how p53 status affects cell migration and invasion in the presence of MLN4924.
- The authors should knockout p53 using CRISPR in cells with wild type p53 and see if it influences the migratory property of the cells treated with MLN4924.
- To prove that Neddylation blockade promotes proteasomal degradation of Slug in cancer cells expressing wildtype p53, the authors should treat SKOV-3 cells expressing wildtype p53 with MG132 in the presence of MLN4924.
- Vimentin is an important mesenchymal marker. Can the authors explain, why there is no change in the expression of vimentin in these cells.
Minor comments:
Description of Figure 6F is missing.
Round 2
Reviewer 3 Report
The authors have addressed all of the reviewer comments. There is no further comment or suggestions for the authors.